# Impact of Environmental and Pharmacologic Changes on the Upper Gastrointestinal Microbiome

**DOI:** 10.3390/biomedicines9060617

**Published:** 2021-05-29

**Authors:** Joshua Bilello, Ikenna Okereke

**Affiliations:** 1Department of Surgery, University of Texas Medical Branch, Galveston, TX 77555, USA; jobilell@utmb.edu; 2Division of Thoracic Surgery, Henry Ford Hospital, Detroit, MI 48202, USA

**Keywords:** microbiome, gastrointestinal tract, proton pump inhibitors

## Abstract

Diseases of the upper gastrointestinal tract have become more prevalent over time. Mechanisms of disease formation are still only partially understood. Recent literature has shown that the surrounding microbiome affects the propensity for disease formation in various parts of the upper gastrointestinal tract. A review was performed of any literature to our best knowledge concerning the effects of pharmacologic agents, environmental changes, and surgical intervention on the microbiome of the upper gastrointestinal tract. Searches of the literature were performed using specific keywords related to drugs, surgical procedures, and environmental factors. Many prescription and nonprescription drugs that are commonly used have varying effects on the upper gastrointestinal tract. Proton pump inhibitors may affect the relative prevalence of some organisms in the lower esophagus and have less effect in the proximal esophagus. Changes in the esophageal microbiome correlate with some esophageal diseases. Drugs that induce weight loss have also been shown to affect the microbiomes of the esophagus and stomach. Common surgical procedures are associated with shifts in the microbial community in the gastrointestinal tract. Environmental factors have been shown to affect the microbiome in the upper gastrointestinal tract, as geographic differences correlate with alterations in the microbiome of the gastrointestinal tract. Understanding the association of environmental and pharmacologic changes on the microbiome of the upper gastrointestinal tract will facilitate treatment plans to reduce morbidity from disease.

## 1. Introduction

The human microbiome contains trillions of microorganisms, many of which are bacteria that assist the host by aiding in the digestion of nutrients and playing a vital role in the innate and adaptive immune system of the host [1]. Microbial dysbiosis, or alteration of the microbiome, plays an important role in chronic inflammation, dysplasia, cancers, and other esophageal diseases [2,3].

The gastrointestinal tract has an important relationship with the microbial community within its lumen. The host–pathogen interaction within the gastrointestinal tract is critical in the development or prevention of disease. Although most literature has focused on the microbiome of the lower gastrointestinal tract and its association with disease, the microbiome of the upper gastrointestinal tract is increasingly being studied [4,5,6].

A review of the literature concerning the upper gastrointestinal tract shows that environmental differences, pharmacologic changes, and specific surgeries can affect the microbiome, which can lead to disease. Our goals are to detail the association of environmental and pharmacologic changes on the upper gastrointestinal microbiome and how those changes may increase the risk of disease.

## 2. Materials and Methods

A detailed review of the literature was performed. Keywords used for the literature review included esophageal microbiome, gastric microbiome, duodenal microbiome, gastroesophageal reflux disease (GERD), eosinophilic esophagitis (EOE), gastric ulcer disease, proton pump inhibitor (PPI), gastric sleeve surgery, Roux-en-Y gastric bypass surgery, gastric ulcer surgery, gastrojejunostomy surgery, Barrett’s esophagus, esophageal squamous cell cancer, esophageal adenocarcinoma, gastric adenocarcinoma, duodenal adenocarcinoma, rural microbiome, metropolitan microbiome, urban microbiome, Gram-positive vs. Gram-negative organisms, and antibiotics. The anatomy of the upper gastrointestinal tract was separated into esophagus, stomach, and duodenum. Environmental and pharmacologic associations with shifts in the microbiome were reported for each section of the upper gastrointestinal tract. Although fecal microbiota transplantation is an important aspect of the treatment of some diseases of the gastrointestinal tract, it was omitted from this discussion of the upper gastrointestinal tract microbiome.

## 3. Results

### 3.1. Esophagus

#### 3.1.1. Differences in Microbiome for Various Esophageal Diseases

The esophageal microbiome has been classified as two types in humans [7]. Type I is composed of Gram-positive bacteria, specifically of the Firmicutes phylum, and is associated with a normal esophagus. Type II is composed of Gram-negative bacteria, including *Veillonella*, *Prevotella*, *Haemophilus*, *Neisseria*, *Granulicatella*, and *Fusobacterium*. The type II esophageal microbiome is more likely to be associated with gastroesophageal reflux disease (GERD) and Barrett’s esophagus. Given the association of Type II with diseases, there is increased interest in the pathogenesis of the esophageal microbiome for this type.

GERD has been clearly linked to the Type II esophageal microbiome and is one of the most common disorders of the gastrointestinal tract, with an estimated prevalence of 25% in the United States [8]. When GERD is left untreated, it can cause bleeding, scarring, and ulcer formation at the gastroesophageal junction, leading to chronic inflammation. This chronic inflammation can lead to a conversion of normal squamous epithelium into metaplastic columnar epithelium of the esophageal mucosa known as “Barrett’s esophagus”. Barrett’s esophagus predisposes patients to the development of dysplasia and is a strong risk factor for the formation of esophageal adenocarcinoma [9,10].

Mechanistic factors in the development of Barrett’s esophagus have been analyzed by comparing the microbiota of gastrointestinal pathologies with controls [11]. These studies have shown that although the overall numbers of bacteria were similar in patients with Barrett’s esophagus compared to control patients, the level of diversity was altered in patients with Barrett’s esophagus [12]. Previously abundant *Streptococcus* species were reduced in patients with Barrett’s esophagus, and the number of Gram-negative anaerobes was increased. Specifically, the genera *Veillonella*, *Prevotella*, *Fusobacterium*, and *Neisseria* were all increased in prevalence in patients with Barrett’s esophagus. This shift from Gram-positive aerobic to Gram-negative anaerobic species may be influenced by environmental changes and related to an abnormal disease state in patients with these pathologies. These findings were further supported by a more recent study [13] which found that patients with esophageal adenocarcinoma had less diverse microbiota compared to control groups.

Differences in diversity included decreased amounts of specific Gram-negative and Gram-positive bacteria, such as *Veillonella* and *Granulicatella*, respectively. However, the Gram-positive bacteria *Lactobacillus fermentum* was found to be increased in esophageal adenocarcinoma patients compared to control patients. *Lactobacillus fermentum*, or other lactic acid-producing bacteria, could dominate and alter the environment of the esophageal mucosa, leading to a low-pH environment and facilitating increased bacterial growth. Fermentation from lactic acid producing organisms could also alter the intraluminal environment and allow *Lactobacillus* to grow preferentially.

The role of oral microbiota in the development of esophageal disease has been studied. Previous studies have shown that the periodontal pathogen *Tannerella forsythia* has been associated with an increased risk of esophageal adenocarcinoma [14]. In addition, the presence of oral *Helicobacter pylori* has been correlated with the presence of gastroesophageal ulcers [15].

The molecular pathways involved in the formation of Barrett’s esophagus have been investigated previously [16,17,18]. Lipopolysaccharides (LPSs), found in the outer membrane of some Gram-negative organisms, can cause activation of toll-like receptor 4 or inflammatory nuclear factor kappa beta (NF-KB) and subsequent expression of numerous proinflammatory cytokines. In addition, shifts in the esophageal microbiome in patients with Barrett’s esophagus have been linked to differential expression of genes responsible for tumor suppression and cell proliferation [19,20,21,22].

While these studies may suggest that microbiome differences lead to pathologies, a previous study [23] posited that the mucosal pathology is the cause of the microbiome shift, rather than a product of the microbiome shift. This position was justified based on results from an experiment that used culture analyses with PCR for specific bacterial taxa and in patients with GERD and Barrett’s esophagus. In this study, an increase in *Campylobacter concisus*, rarely found in esophageal biofilm, was discovered. It was hypothesized that the presence of this bacteria in the esophagus is from chronic reflux and is a consequence of GERD. They also found a significant increase in the expression of interleukin-18, a proinflammatory cytokine that is known to induce interferon-y and plays an important role in host immunity in patients with GERD and Barrett’s esophagus. The study concluded that there may be a possible relationship between the esophageal microbiome and inflammatory markers triggered by certain alterations in the microbiome secondary to diseases.

Another pathology potentially linked to differences in esophageal microbiomes is EOE, which is found in pediatric and adult populations with a reported incidence of 0.1–1.2 per 10,000 people worldwide [24]. Little is known about the role of the esophageal microbiome in those with EOE. Microenvironmental factors such as the microbiome are now being studied, however. EOE occurs due to an immunogenic reaction to antigens that are commonly found in food and air pollution. These factors trigger a Th2-type response from the host, resulting in infiltration of eosinophils into the esophageal mucosa. Previous studies have shown a significant increase in Gram-negative organisms such as *Haemophilus* in patients with EOE compared to control [25]. These studies also found that the bacterial load was increased in patients with EOE compared to control subjects. In fact, the amount of bacteria in patients with EOE was increased regardless of treatment status or the degree of eosinophilia when compared with control patients. Furthermore, another study [26] found an increase in Gram-negative *Neisseria* in those with EOE compared to control subjects. An increase in Gram-positive *Corynebacterium* was also discovered in this study. Interestingly, this study also showed that the oral microbiome was distinct in patients with EOE compared to patients without EOE. The increase in both *Neisseria* and *Corynebacterium* suggests that these two organisms may have increased absolute levels secondary to inflammation and not EOE itself. However, given the limited studies on EOE and esophageal microbiomes, future studies are necessary to explicate confounds in patients with EOE. Although intuitive, there is not a consensus yet regarding pollutants found in air and EOE. Some investigators have found no correlation, in fact [27].

Achalasia, a motility disorder of the lower esophagus, is signified by dysphagia, regurgitation of food, and weight loss due to the inability of the lower esophageal sphincter to relax. Achalasia and the esophageal microbiome also have not been well evaluated to date. There have been several case reports that show an association between primary and secondary achalasia due to HIV and an increase in *Mycobacterium 3oodie* [28,29].

While specific microbiome differences may exist for certain pathologies, the outcomes of the previous studies demonstrate the exploratory nature of current research and the need for studies that evaluate the differences in microbiomes to further understand their pathogenesis.

#### 3.1.2. Effect of Proton Pump Inhibitors

PPIs are one of the most widely used classes of drugs in the United States and have also been hypothesized to alter the esophageal microbiota. PPIs suppress gastric acid production by increasing the pH of the stomach. The pH is increased secondary to inhibition of the H+/K+ ATPase receptor, which is responsible for secreting acid into the gastric lumen. Suppression of gastric acid by PPIs can cause bacterial overgrowth and alterations in the esophageal microbiome [30].

PPI use has been associated with a number of alterations, including in a study [19] that showed that PPIs led to an altered microbiota pattern and decreased microflora diversity. In this study, PPI use was associated with an increase in *Streptococcaceae* and other lactobacilli. Other significant changes after PPI treatment include significantly decreased levels of *Comamonadaceae*, *Proteobacteria*, and *Bacteroidetes* and significantly increased levels of *Clostridiaceae*, *Lachnospiraceae*, *Micrococcaceae*, *Actinomycetaceae*, *Lactobacillales*, *Gemellales*, *Clostridia*, and bacteria of the *Firmicutes* phylum [31,32]

These results suggest a more neutral environment secondary to PPI use is favorable for certain bacteria to thrive. However, mixed findings within these studies should prompt future research to clarify these results. Some studies have even shown no change in certain microbiota with PPI use (Figure 1a,b) [33].

#### 3.1.3. Effect of Bariatric Surgery

One of the most effective weight loss strategies for obesity is surgical intervention. The Roux-en-Y Gastric Bypass (RYGB) is the most common and effective bariatric surgery performed in the United States and can facilitate a tremendous amount of weight loss in an individual. After the procedure, the digestive system and metabolism of the gastrointestinal tract are affected. Portions of the esophagus, stomach, and intestine are modified, which alters the bacterial populations in each of those areas. As these surgeries may alter the amount of acidic reflux into the esophagus, the microbiota of the distal esophagus may be altered following surgery.

#### 3.1.4. Effect of Antibiotics

Antibiotics can affect the esophageal microbiome through indirect and direct mechanisms of action. Antibiotics are administered to patients to kill harmful bacteria, but due to their broad spectrum of activity, they may kill potentially beneficial organisms in the process. They may also affect the esophageal microbiome in an indirect manner by disrupting the homeostasis of the entire microbiome, as many organisms rely on secondary metabolites produced by certain parts of the microbiome to function. An analysis of the effect of antibiotic use showed that there were alterations in the microbiome related to antibiotics. This same study showed no relationship between these changes and the development of esophageal adenocarcinoma, however. These results suggest that the esophageal microbiome may not be correlated to the formation of esophageal adenocarcinoma.

Results from another study that analyzed the microbiome in patients treated or not treated for *Helicobacter pylori* infection showed that the number of species in the distal esophagus was significantly reduced, especially *Lactobacillales*, in the treatment group [34]. The number of Gram-negative bacteria was not increased, but the colonization of *Staphylococcus*, *Acinetobacter*, and nonspore *Bacillus* was increased.

The molecular mechanisms for *Helicobacter pylori* colonization and survival in the upper gastrointestinal tract after antibiotic administration have been studied [35,36,37]. While the amount in the distal esophagus is reduced, some strains of *Helicobacter pylori* persist or even increase following antibiotic administration. One of the main molecular mechanisms by which *Helicobacter pylori* develops resistance to some antibiotics is with a point mutation of domain V of the 23S ribosomal rRNA gene. This point mutation results in the inability of clarithromycin to bind to the 50S ribosomal subunit, thereby limiting the effectiveness of clarithromycin. Fluoroquinolone resistance is mediated by a mutation in the gene that produces gyrA. Amino acid positions 87 and 91 are altered in the mutated form, making *Helicobacter pylori* less sensitive to fluoroquinolone treatment.

It has been suggested that antibiotics could potentially be used in the chemoprevention of esophageal adenocarcinoma by converting a Type II microbiome to a Type I microbiome by increasing the abundance of Streptococcus [38,39,40]. More research needs to be performed to determine the specific role of antibiotics in altering the esophageal microbiome [41]. Figure 2 shows the effects of antibiotics on specific organisms throughout the upper gastrointestinal tract.

#### 3.1.5. Impact of Environmental Differences (i.e., Rural vs. Urban)

There are significant disparities and variations in the diets of rural versus urban populations [42]. The diets of rural and urban populations can differ based on the availability of foods and the socioeconomic status of individuals living within a community. People living in more urban areas have an increased exposure to diets rich in simple sugars, animal proteins, and fats since these foods are more accessible and cheaper than healthier options. People in rural communities, in contrast, tend to have a diet that contains more fiber than those living in urban areas. These stark contrasts in diets can have a significant effect on the microbiota. In addition, there are vast differences in the environmental and social factors that people face in urban versus rural communities. There are significant differences in employment rate, access to healthcare, levels of pollution, and level of stress between these communities. The racial makeup also tends to be different between these communities. These factors have been increasingly studied.

A study found that the gut microbiome in children living in a village in rural Africa was vastly different than the microbiome of children living in the Western world [43]. It was hypothesized that children in rural Africa ate a diet richer in fiber compared to children with a typical Western diet rich in fats, proteins, and simple sugar. This difference in diet between the two sets of children had a significant effect on their esophageal microbiomes. Rural African children showed a significant increase in *Bacteroidetes* and a decrease in *Firmicutes*. They also had an increase in *Prevotella* and *Xylanibacter*. *Enterobacteriaceae* were significantly lower in rural African children compared to Western children. The authors hypothesized that the microbiome has evolved over time in conjunction with the diets of these individuals to allow for maximum energy from their specific diets.

There have been studies examining the molecular basis by which dietary and environmental alterations lead to shifts in the gut microbiome and esophageal disease [44,45,46]. Mice fed with a high-fat diet were found to have increased production of IL-8 compared to control mice. These increased IL-8 levels were associated with increased development of dysplasia. When 16S rRNA cluster analysis was performed, the mice with the high-fat diet clustered separately from control mice and had an altered beta diversity level.

Other environmental factors, such as exposure to pollutants and poor air quality, have also been seen to affect the gut microbiome [47,48]. Poor drinking water, for example, has been seen to increase the production of N-nitroso metabolites from gut organisms. Specific contaminants in water may lead to an increased level of these N-nitroso compounds [49].

### 3.2. Stomach

#### 3.2.1. Differences in Microbiome for Various Diseases of the Stomach

Previous research on the gastric microbiota was less extensive for many years because many investigators believed that the acidic environment of the stomach was not hospitable for most organisms. However, the discovery of *Helicobacter pylori* by Robin Warren and Barry Marshall in 1982 led to a new theory that bacteria were capable of colonizing the entire gastrointestinal tract. Current research supports this hypothesis, and recent results suggest that the stomach is home to a diverse microbiome including *Prevotella*, *Streptococcus*, *Veillonella*, *Rothia*, *Helicobacter*, and *Haemophilus*. Research has also shown that certain gastric diseases correlate with alterations in the gastric microbiome [50,51,52]. *Helicobacter pylori* infections are responsible for many gastric diseases and are an established risk factor for gastric cancer. Alterations in the gastric microbiome are closely tied to *Helicobacter pylori* status in an individual, and many studies have shown the relationship between this bacteria and gastric pathology. It also appears that the titer of *Helicobacter pylori* may be associated with the development of gastric ulcers and gastric cancer [53].

Chronic gastritis, defined as inflammation of the central aspect of the stomach, is very common in the world. An experiment sampled the gastric mucosa of patients with chronic gastritis with 16S rRNA gene sequencing and found an abundance of *Prevotella*, *Streptococcus*, *Neisseria*, *Porphyromonas*, and *Haemophilus* compared to controls [54]. In *Helicobacter pylori*-infected individuals with gastritis, another study found that the level of Proteobacteria was decreased while the level of Firmicutes was increased compared to *Helicobacter pylori*-negative individuals. This pattern suggests *Helicobacter pylori* infection contributes to the microbiome of the stomach [55]. Furthermore, in patients with confirmed atrophic gastritis, the levels of *Streptococcus* were increased while *Prevotella* was decreased when compared with healthy controls [56]. Chronic gastritis patients also had a higher rate of bacterial growth compared to controls, suggesting that *Helicobacter pylori* is not the sole component of gastritis and that other bacteria may play a role [57]. The changes in microbiome may also be related to changes in pH, although further studies are needed to examine this relationship.

More specifically, this theory applies to antral gastritis as well. One study [58] found that patients with antral gastritis and *Helicobacter pylori* infection had a relative decrease in *Proteobacteria* and *Prevotella* and an increase in *Firmicutes* and *Streptococcus* when compared to patients without *Helicobacter pylori* infection. They also found a significant increase in *Streptococcus* species may lead to antral gastritis. This further supports the theory that alterations in the gastric microbiome are capable of causing disease states in the stomach.

Gastric biopsies taken from individuals with peptic ulcer disease with identified *Helicobacter pylori* infections were found to have increased levels of *Streptococcus*, *Neisseria*, *Rothia*, and *Staphylococcus* via mass spectrometry. These bacteria are known to be more apt to grow and thrive in a low-pH environment. Hu and colleagues found that healthy controls had more predominantly acid-resistant microbes compared to patients with peptic ulcers. Those with confirmed *Helicobacter pylori* infection with gastric ulcers were found to have a much lower level of organisms other than *Helicobacter pylori* in comparison to individuals with identified non-ulcer-related dyspepsia [59].

*Helicobacter pylori* infection is also an established risk factor for gastric cancer, and treatment for this infection has been shown to decrease rates of gastric adenocarcinoma in infected individuals. Coker and colleagues [60] found that there was a higher abundance of oral bacteria in patients with gastric cancer when compared to patients with chronic atrophic gastritis or intestinal metaplasia. *Peptostreptococcus stomatis*, *Slackia exigua*, *Parvimonas micra*, *Streptococcus anginosus*, and *Dialister pneumosintes* were the most abundant organisms. Eun and colleagues [61] proposed that an alteration of the entire gastric microbiome could play a role in the pathogenesis of gastric cancer. Another study [62] found that antibiotic treatment for the eradication of *Helicobacter pylori* led to an increase in Cyanobacteria, Bacteroidetes, Fusobacteria, and Actinobacteria and a decrease in *Proteobacteria*, *Epsilonproteobacteria*, *Campylobacterales*, *Helicobacteraceae*, *and Helicobacter*.

Jimenez and colleagues [39] found that the bacterial microbiome tends to have decreased diversity levels in patients with invasive gastric cancer compared to patients without cancer. There was a relative decrease in *Porphyromonas*, *Neisseria,* and *Streptococcus sinensis*, while there was an increase in *Lactobacillus coleohomonis* and *Lachnospiraceae*. *Pseudomonas* was significantly more abundant in patients with gastric cancer than in patients with non-atrophic gastritis. A shift in these altered levels of bacteria has been proposed to favor the development of gastric cancer.

Numerous studies have attempted to characterize the molecular pathways that are involved in the development of gastric disease from shifts in the microbiome [63,64,65,66,67,68,69,70]. One study performed biopsies in patients with gastric cancer (*n* = 10) and no disease (*n* = 5) [71]. Terminal restriction fragment length polymorphism (T-RFLP) analysis was used in combination with 16S rRNA gene sequencing to analyze the microbiome differences in patients with and without gastric cancer. Out of the 140 sequenced clones, 102 phylotypes were found. Five different clusters were identified. *Helicobacter pylori* was found in only one of the clusters. Forty-nine distinct terminal restriction fragments were found, with each patient having a mean of seven terminal restriction fragments. Interestingly, none of the distinct terminal restriction fragments were common among the cancer patients. In this study, the diversity indices did not differ between cancer and control patients. But four out of the five healthy patients clustered together, suggesting that there were more similar bacterial communities in healthy patients than in patients with cancer.

Another proposed mechanism by which the microbial community can cause gastric cancer is by the production of N-nitroso compounds. Some organisms can produce N-nitroso compounds, some of which have been verified to be carcinogens. Several studies have hypothesized that shifts in the microbiome toward organisms capable of producing N-nitroso compounds increase the risk of gastric cancer [72,73,74,75].

Gastric microbiota may also induce gastric cancer by inducing oxidative stress, genotoxicity, and chronic inflammation [76,77]. Activation of NF-kB by some organisms has been associated with an increased risk of cancer development [78,79,80].

According to the studies shown above, the gastric microbiome is significantly altered in many disease states of the stomach. *Helicobacter pylori* is a key inciting factor in the pathogenesis of many of these diseases, but other shifts in the microbiome can contribute to gastric diseases as well. Further research must be done to establish if alterations in this microbiome are the pathogenesis of gastric disease or if they are a consequence of the disease states themselves.

#### 3.2.2. Effect of Proton Pump Inhibitors

The use of PPIs has been shown to cause alterations throughout the gastrointestinal tract, including the gastric microbiome. PPIs have been found to alter the gastric microbiome composition and increase the diversity of the microbiome compared to controls in several recent studies. PPIs work directly at the gastric mucosa on the H+/K+ ATPase pumps. Alterations in the gastric microbiome are believed to be due to a significant increase in pH that is secondary to PPI use. However, other mechanisms could be involved [81]. The current leading theory states that the gastric microbiome is disrupted secondary to PPI use and that direct targeting of bacterial and fungal proton pumps is how PPIs affect the gastric bacterial composition [41].

Gastric fluids sampled from individuals who were on PPI medications were found to be significantly different compared with controls [82]. Levels of *Moraxellaceae*, *Flavobacteriaceae*, *Comamonadaceae*, and *Methylobacteriaceae* were significantly decreased, while *Erysipelotrichaceae* was increased. Two hypotheses have been proposed for these alterations in the gastric fluid microbiome: (1) This alteration is secondary to the increase in pH caused by PPI therapy, which allows bacteria that prefer a more alkaline environment to thrive compared to the more acidic bacteria of the average stomach. (2) This alteration is due to host-mediated effects secondary to PPI use.

These alterations in the gastric microbiome are potentially linked with an increased risk for *Clostridium difficile*. Additional studies are needed, however, to determine the mechanism behind the association of prolonged PPI use and Clostridium infection. The question of whether alterations due to PPIs are beneficial or harmful to the gastric microbiome requires further study. Moreover, some patients with EOE respond to PPI administration. Future studies are required to determine whether the microbiota of EOE patients who respond to PPI use differs from the microbiota of EOE patients who do not respond.

Some studies have shown that chronic PPI use may be associated with gastric cancer development [83,84,85]. One proposed mechanism is hypergastrinemia which results from PPI use [86,87]. Chronic PPI use may also increase the relative and absolute abundance of organisms that act through these molecular mechanisms to cause cancer development [88]. PPI use also increases the nitrate/nitrite reductase activity of some organisms, which may be involved with cancer development [89,90,91].

#### 3.2.3. Effect of Bariatric Surgery

Gastric bypass surgeries result in rapid weight loss, reduced peripheral adipose tissue, and improved glucose metabolism. There are several types of bariatric surgery, including Roux-en-Y gastric bypass (RYGB), sleeve gastrectomy (SG), and bilio-intestinal bypass (BIB). However, the mechanism behind the metabolic consequences of these operations remains largely unknown. Several studies have shown that bariatric procedures alter the gastric microbiome, which may have an explanatory relationship with the aforementioned metabolic outcomes. Possible mechanisms for the changes in the intestinal microbiota include food choice and preferences, reduction of food consumption, and nutrient malabsorption. Another leading theory is that the effect of these surgeries is due to an altered microbiome interaction. These surgeries result in anatomical and functional modifications of the gastrointestinal tract and have been shown to alter the microbiome of the stomach.

There are also alterations in the molecular mechanisms of the stomach in patients who undergo bariatric surgery. It is unclear whether the alterations are a function of the surgery or the subsequent weight loss experienced by most patients. After surgery, there is a decrease in circulating monocytes, which may affect the amount of inflammation intraluminally [92].

Patients who underwent an RYGB procedure saw an increase in the abundance of bacteria that were mainly from the oral tract, such as *Fusobacteria*, *Veillonella*, and *Greanucatiella* (Figure 3). These facultative anaerobes were likely increased after surgery due to the presence of increased intraluminal oxygen levels occurring after the anatomic alterations that occur with surgery.

Steinert and colleagues [93] found that *Proteobacteria* was increased in those who underwent RYGB compared to control patients. This increase might be secondary to an increase in oxygen availability in the large intestine after surgery, which favors anaerobes such as *Escherichia*, *Klebsiella*, and *Pseudomonas*. A decrease in obligate anaerobic Gram-positive bacteria such as *Blautia*, *Roseburia*, *Faecalibacterium*, and *Bifidobacterium* was found among patients who underwent an RYGB procedure. It is also possible that these changes are linked to reduced gastric acid secretion after RYGB surgery and reduction in total energy intake.

Lu and colleagues [94] found that patients who underwent RYGB and SG procedures had decreased serum uric acid, interleukin-6, tumor necrosis factor alpha, LPS, and xanthine oxidoreductase (XO) activity. Obesity has been shown to be associated with increased levels of XO, uric acid, and cytokines. These procedures also were shown to alter the diversity of the gastric microbiome significantly. Compared to the control group, patients who underwent RYGB or SG procedures were found to have increased abundance of *Verrucomicrobia* and *Akkermansia muciniphila*, while *E. coli* levels were decreased. The decreased abundance of *E. coli* after RYGN and SG demonstrates that the alteration in *E. coli* may be a factor that regulates XO expression by influencing LPS levels.

Patients undergoing SG surgery had an increase in *Bacteroidetes* and a reduction in *Firmicutes*. Machado and colleagues [95] studied samples at three and six months after surgery and found that the levels of *Clostridium*, *Eubacterium*, *Faecalibacterium*, *Dorea*, and *Coprococcus* had significantly declined after surgery. Sanmiguel and colleagues found that there was a significant decrease in *Bifidobacteriaceae* and an increase in *Fusobacterium*, *Atopobium*, and *Bulledia* [96].

Patients who underwent a BIB procedure were found to have similar alterations in the gastric microbiome compared to other weight-loss surgeries [97]. Decreased levels of *Lachnospiraceae*, *Clostridiaceae*, *Ruminococcaceae*, *Eubacteriaceae*, *Coriobacteriaceae*, and *Carnobacteriaceae* were identified at six months after the procedure in those who received BIB compared to controls. The genera of note that were reduced were *Faecalibacterium*, *Ruminococcus*, *Clostridium*, *Eubacterium*, and *Blautia*. However, levels of *Megasphaera*, *Acidaminococcus*, *Lactobacillus,* and *Enterobacteriaceae* were all significantly increased in patients who underwent the BIB procedure. These preliminary findings suggest that there are similarities in the alterations of the gastric microbiomes, but there is limited evidence detailing the impact of BIB procedures on the gastric microbiome, and more research must be done to confirm the findings above.

In all three bariatric surgeries mentioned above, there was a significant change in the gastric microbiome following surgery. The exact correlation between alterations in the microbiome and bariatric surgery remains a mystery and may provide new therapeutic options in addition to procedures, such as the RYGB, in combating morbid obesity. There are several studies that have proposed beneficial effects of common probiotics for bariatric surgery patients [98,99]. Moreover, dietary changes following bariatric surgery may affect the microbiome of the gastrointestinal tract.

### 3.3. Duodenum

#### 3.3.1. Differences in Microbiome with Duodenal Ulcer Disease

Small intestinal bacterial overgrowth (SIBO) is characterized by an excessive number of bacteria in the small intestine and is associated with symptoms such as bloating, abdominal discomfort, diarrhea, and weight loss [100]. SIBO has been known to play an important role in other pathologic GI diseases such as inflammatory bowel disease (IBD) and fatty liver. Understanding the microbiome composition of individuals with SIBO is an important step in understanding the pathophysiology behind SIBO and other common GI pathologies. The microbiome of patients with SIBO was found to have an increase in the abundance of Proteobacteria and a decrease in *Firmicutes*, which are a normal component of the small intestinal gut flora. The increase in abundance of Proteobacteria is also coupled with the fact that the composition of the phylum was significantly altered in those with SIBO. There was an increase in *Gammaproteobacteria*, *Enterobacteriaceae*, and *Aeromonas* and a decrease in *Alphaproteobacteria* in those with SIBO. These findings are consistent with previous studies that the bacterial overgrowth in SIBO could be due to organisms found in the colon.

The microbiota profile of those with celiac disease, when compared with patients without disease, exhibited an increased abundance of *Prevotella* [101]. *Serratia* was also found to be present in higher amounts in patients with celiac disease. These bacteria may impair the intestinal integrity in patients with celiac disease, but their role in the pathogenesis remains unknown. One potential molecular mechanism to explain the role of microbiome alterations in the development of celiac disease was recently proposed. A study of 20 adults with celiac disease revealed genetic diversity of the iron acquisition systems and some hemoglobin-related genes [102]. In addition, a diet low in gluten has been shown to create changes in the microbiome of the gastrointestinal tract [103].

A study [104] on children with ulcerative colitis found that they have a significantly altered composition of their duodenal microbiome. Children with ulcerative colitis were found to have low levels of *Actinobacteria*, *Bacteroidetes*, and *Firmicutes* compared to non-IBD controls. This alteration in the duodenal microbiome could be due to local inflammation of the intestinal mucosa. This study highlights that alterations in the microbiome are not just limited to the colon in ulcerative colitis but may extend throughout the small intestine [105,106,107].

Significant differences were found in the duodenal microbiome in duodenal mucosa between children with Crohn’s disease and controls [108]. Schmitt and colleagues found that there was a significant increase in *Pseudomonadales* in controls while there was an abundant increase in *Prevotellaceae* in those with Crohn’s disease. This confirmed the previously established model that IBD significantly alters the small intestinal microbiome, including in those with Crohn’s disease.

Saffouri and colleagues [109] found that the duodenal microbiome was altered in patients with common gastrointestinal symptoms such as diarrhea, abdominal pain, and bloating. The duodenal microbiome in those with symptoms showed less alpha diversity along with a decrease in *Porphyromonas*, *Prevotella*, and *Fusobacterium*. Symptomatic patients also showed a higher level of heterogeneity of the gastric microbiome when compared with controls. Patients’ advanced age, antibiotic and PPI use [110], and history of GI surgery make contributions to alterations in the duodenal microbiome.

Many GI diseases alter the microbiome of the duodenum, causing intestinal dysbiosis [111]. More research must be done to open new possibilities of treatment options for patients with chronic diseases affecting the duodenum.

#### 3.3.2. Effect of Proton Pump Inhibitors

PPIs have been shown to affect the duodenum by a mechanism similar to that in the esophagus and stomach. A study that examined the effects of PPI use on the microbiome of the small intestine and stool showed no differences in most organisms in the duodenal microbiome with PPI use at the phylum, class, or order levels [112]. However, PPI use was associated with an increase in *Campylobacteraceae* and a significant decrease in *Clostridiaceae*. These results suggest that the effect of PPIs on the duodenal microbiome is minimal. However, the clinical importance of the increased abundance of *Campylobacteraceae* and *Clostridiaceae* remains unknown. In addition, the interaction of nonsteroidal medications and PPIs remains fully unknown [113].

### 3.4. Future Directions

There will be significant research in the future on the mechanisms of action of the microbiome on disease in the gastrointestinal tract. While there have been many studies that have analyzed the microbiome in a series of patients, future studies will be able to perform ex vivo and animal studies. The advantage of these models is that there will be the ability to standardize the environment and introduce specific organisms. Studies that have looked at hundreds of patients with esophageal cancer, for example, cannot completely control factors such as diet, weight loss, and geographic location. All of these factors may skew the results of their analyses of the microbiome of the gastrointestinal tract. However, performing these experiments in animals may allow for better standardization.

In the future, there will also be more investigation into the effects of chemotherapy on the microbiota of the upper gastrointestinal tract. Most previous studies have focused on the changes in the lower gastrointestinal tract. There is some evidence, however, that patients receiving chemotherapy may experience significant decreases in some organisms that may affect the gut–brain axis [114].

In particular, the use of germ-free or gnotobiotic rodents will provide an opportunity for investigators to test hypotheses about organism exposure and its relation to disease development. There are now reliable protocols that can generate gnotobiotic rodents and allow for experimentation with a standardized microbiome [115]. In the future, experiments will be performed to find causative mechanisms of action of particular organisms in creating disease.

The discovery of probiotic organisms would have significant implications for treatment. Future trials may explore the role of oral supplementation in preventing disease, especially in high-risk cohorts. Patients at high risk for esophageal cancer, for example, may be shown to benefit from a diet supplemented with a probiotic organism. Screening tests, which now rely on visual examination and histologic analysis, may also measure the microbiome during the screen. In the future, the microbiome analysis will likely be used to determine the treatment algorithm for patients with gastrointestinal disease.

### 3.5. Conclusions

The microbiome of the gastrointestinal tract is being increasingly analyzed. The microbiome varies significantly in different parts of the upper gastrointestinal tract. Changes in the microbial community at each of these locations have been associated with disease. Future experiments will utilize standardized environments, using either ex vivo or animal models, to determine the effects of specific organisms on the development of disease. In the future, treatment of patients will likely be affected by the microbiome analysis of individual patients.

## Figures and Tables

**Figure 1 biomedicines-09-00617-f001:**
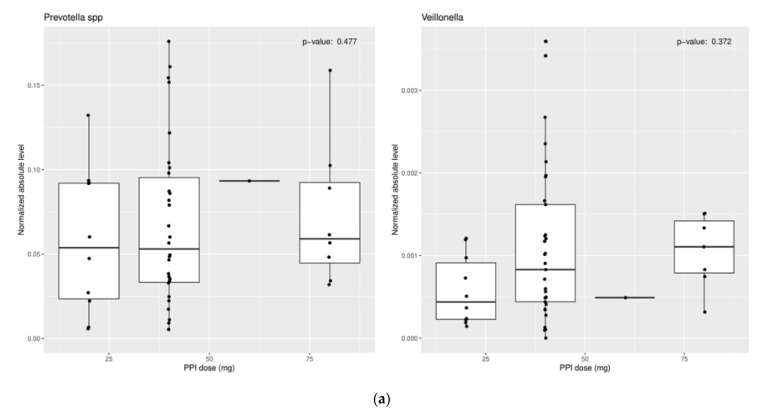
Relationship of (**a**) PPI dose and (**b**) PPI duration of use vs. normalized absolute levels of organism for Prevotella and Veillonella. There was no association between PPI dose or duration of use and organism level.

**Figure 2 biomedicines-09-00617-f002:**
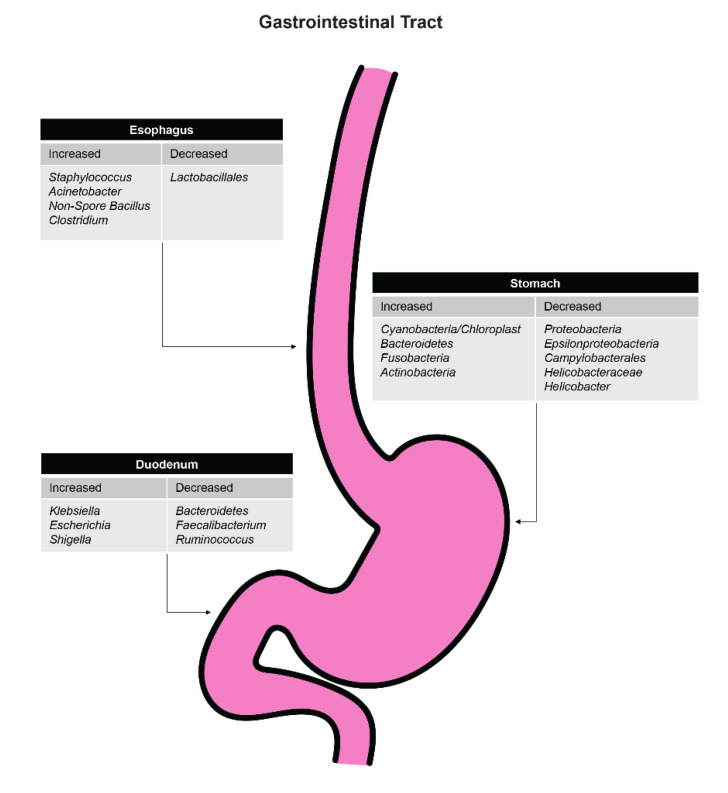
Effects of antibiotics on organism levels in various aspects of the upper gastrointestinal tract.

**Figure 3 biomedicines-09-00617-f003:**
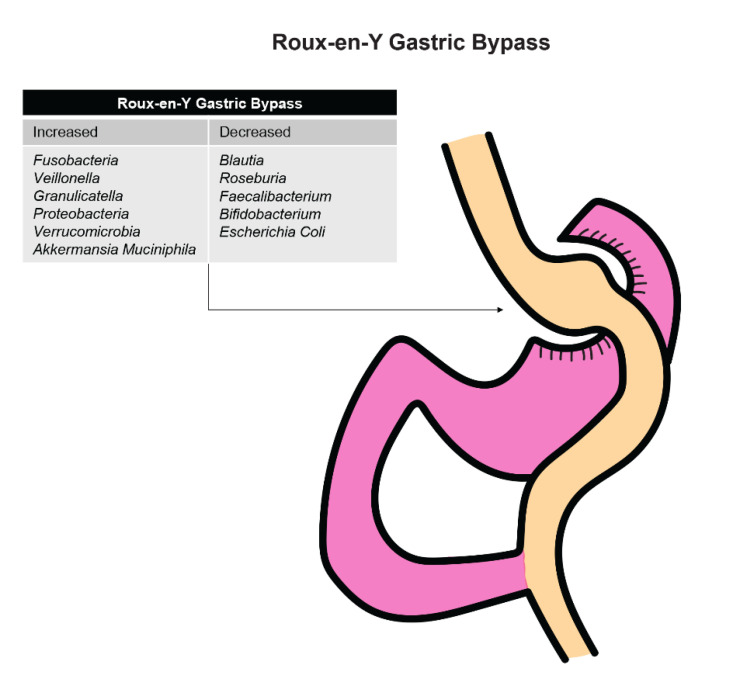
Effects of Roux-en-Y gastric bypass on organism levels in the stomach.

## Data Availability

The data presented in this study are available on request from the corresponding author. The data are not publicly available due to privacy concerns. All figures in this manuscript were created by the authors and are also available upon request.

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
