# Peer review of "Impact of Environmental and Pharmacologic Changes on the Upper Gastrointestinal Microbiome"

_biomedicines, 2021, doi:10.3390/biomedicines9060617_

Round 1

Reviewer 1 Report

In general,the paper is in good standard but the total omission of the discussion/conclusions/summary/future directions is my major consern. Otherwise this paper would have passed my review with minor revision.

Line 11

“all literature concerning” -> this is a bold statement. Should have perhaps “to our best knowledge” added

Line 32

Dysbiosis has been suspected and demonstrated well before 2019. This paper should be cited as it is one of the landmark papers>

Mazmanian SK, Round JL, Kasper DL. A microbial symbiosis factor prevents intestinal inflammatory disease. Nature. 2008 May 29;453(7195):620-5. doi: 10.1038/nature07008. PMID: 18509436.

Line 37

UGIT has been suspected and demonstrated well before 2016. This paper should be cited as it is one of the landmark papers>

Macfarlane S, Furrie E, Macfarlane GT, Dillon JF. Microbial colonization of the upper gastrointestinal tract in patients with Barrett's esophagus. Clin Infect Dis. 2007 Jul 1;45(1):29-38. doi: 10.1086/518578. Epub 2007 May 22. PMID: 17554697.

Line 45-53

Check the presence of extra spaces before start of the sentences. Also, on line 75, 120, 320 and 421.

Line 55

It should be noted that some aspects were left outside this review, one of the most important ones is FMT (Fecal Microbiota Transplantation). This omission is fine as long as it is stated in the aims/scope, i.e in this section.

Line 99

A sentence should not start with “And”. Rephrase.

Line 101

Instead of reviews, original research should be sited here. A couple of selected from here perhaps>

  • Greenawalt DM, Duong C, Smyth GK, Ciavarella ML, Thompson NJ, Tiang T, Murray WK, Thomas RJ, Phillips WA. Gene expression profiling of esophageal cancer: comparative analysis of Barrett's esophagus, adenocarcinoma, and squamous cell carcinoma. Int J Cancer. 2007 May 1;120(9):1914-21. doi: 10.1002/ijc.22501. PMID: 17236199.
  • Parenti A, Leo G, Porzionato A, Zaninotto G, Rosato A, Ninfo V. Expression of survivin, p53, and caspase 3 in Barrett's esophagus carcinogenesis. Hum Pathol. 2006 Jan;37(1):16-22. doi: 10.1016/j.humpath.2005.10.003. PMID: 16360411.
  • Jankowski J, Hopwood D, Wormsley KG. Expression of epidermal growth factor, transforming growth factor alpha and their receptor in gastro-oesophageal diseases. Dig Dis. 1993;11(1):1-11. doi: 10.1159/000171396. PMID: 8443952.
  • Klump B, Hsieh CJ, Holzmann K, Borchard F, Gaco V, Greschniok A, Eckardt VF, Bettendorf U, Gregor M, Porschen R. Diagnostic significance of nuclear p53 expression in the surveillance of Barrett's esophagus--a longitudinal study. Z Gastroenterol. 1999 Oct;37(10):1005-11. PMID: 10549095.

Line 103

“a recent study”. This refers to paper from 2013 which is over 8 years old.

Line 104, 106, 107, 110, and 111

In short space we have five (5) sentences starting with “They”. Rephrase using “In this study” or something similar to avoid tautology.

Line 119-120

Although logical, there is not consensus regarding air as yet. So this notion should be mentioned.

Guajardo JR, Zegarra-Bustamante MA, Brooks EG. Does Aeroallergen Sensitization Cause or Contribute to Eosinophilic Esophagitis? Clin Rev Allergy Immunol. 2018 Aug;55(1):65-69. doi: 10.1007/s12016-018-8671-6. PMID: 29356936.

Line 209

“…in a village in rural Africa,…”. The study cited here (40) has looked urban-rural differences in West Java, Indonesia. I could not find anything about Africa! Maybe a mixuo with reference 46?

Line 226-227

This sentence is bit non-informative as this topic is quite complicated and related to dysbiosis etc. Thus, this sentence would benefit at least from one citation>

Kobayashi J. Effect of diet and gut environment on the gastrointestinal formation of N-nitroso compounds: A review. Nitric Oxide. 2018 Feb 28;73:66-73. doi: 10.1016/j.niox.2017.06.001. Epub 2017 Jun 3. PMID: 28587887.

This could be cited as “For a review, see [ ]”

Line 237

The reference given is not relevant for the sentence. These two would be fairly recent reviews unless original research is cited instead.

Krishnareddy S. The Microbiome in Celiac Disease. Gastroenterol Clin North Am. 2019 Mar;48(1):115-126. doi: 10.1016/j.gtc.2018.09.008. Epub 2018 Dec 13. PMID: 30711204.

Chey WD, Kurlander J, Eswaran S. Irritable bowel syndrome: a clinical review. JAMA. 2015 Mar 3;313(9):949-58. doi: 10.1001/jama.2015.0954. PMID: 25734736.

Line 301

Should say “N-nitroso compounds”.  Also, not all of these are verified carcinogens so this sentence should be rephrased accordingly.

Line 303

For clarity, “these” could be deleted and replaced with “organisms capable of producing …”

Line 305

The following comprehensive review could be given>

Wang LL, Yu XJ, Zhan SH, Jia SJ, Tian ZB, Dong QJ. Participation of microbiota in the development of gastric cancer. World J Gastroenterol. 2014 May 7;20(17):4948-52. doi: 10.3748/wjg.v20.i17.4948. PMID: 24803806; PMCID: PMC4009526.

Line 339

“organsims”. Should be “organisms”

Line 345

“behind these” ->  “behind the metabolic outcomes of these operations”

Line 353

“There are” -> “There are also”

Line 400

There are at least half a dozen papers suggesting positive effects of commonly used probiotics to microbiota for bariatric surgery patients so perhaps this should be mentioned at least briefly. For example>

Woodard GA, Encarnacion B, Downey JR, Peraza J, Chong K, Hernandez-Boussard T, Morton JM. Probiotics improve outcomes after Roux-en-Y gastric bypass surgery: a prospective randomized trial. J Gastrointest Surg. 2009 Jul;13(7):1198-204. doi: 10.1007/s11605-009-0891-x. Epub 2009 Apr 18. PMID: 19381735.

Sherf-Dagan S, Zelber-Sagi S, Zilberman-Schapira G, Webb M, Buch A, Keidar A, Raziel A, Sakran N, Goitein D, Goldenberg N, Mahdi JA, Pevsner-Fischer M, Zmora N, Dori-Bachash M, Segal E, Elinav E, Shibolet O. Probiotics administration following sleeve gastrectomy surgery: a randomized double-blind trial. Int J Obes (Lond). 2018 Feb;42(2):147-155. doi: 10.1038/ijo.2017.210. Epub 2017 Aug 30. PMID: 28852205.

Line 448

The paragraph title is not in italics. Check the formatting for consistent presentation of the paper.

Line 456

A sentence should not start with “And”.

Line 457 onwards

Expected some sort of summary or discussion, the paper stops very abruptly. This should be in the same style as other papers in the same journal.

Page 15, Figure 1.

The original source should be mentioned and if the figures we redrawn or reprinted with permission.

Page 16, Figure 2.

The original source should be mentioned and if the figures we redrawn or reprinted with permission.

Page 16, Figure 3.

The original source should be mentioned and if the figures we redrawn or reprinted with permission.

Reviewer 2 Report

This is a well written review, but I have some comments and suggestion to the authors:

  • I'd like the authors write about the role of oral microbiota in esophagous, stomach and duodenum ulcers. Add a comment on it. Think about gingivitis, tooth decay, teeth eruption, etc.
  • Please consider adding information about the changes in microbiota in those patients under chemotherapy treatment.
  • Please add a comment about changes in esophageal microbiota in those patients with Barret's esophagous after surgery. Make a comment
  • Is there any information about correlation of nasal and oral microbiota with Esophageal microbiota? Make a comment.
  • Some patients with EEo respond to PPIs. Is there any diferrence in esophageal microbiota in those group of patientes who respond to PPIs. Make a comment.
  • Regarding H pylori: is there any difference in gastric microbiota in those patients with disease related to H pylori and those with only colonization with H pylori.
  • Line 194: what's the meaning of ECA?
  • Please make a comment of oral microbiota and EEo
  • Line 199 and ongoing: This part must be improved. Rural and urban are not only differenced by food. Please, make a comment about polution, stress, etc.
  • Lines 249-251: what about the pH in those patients.
  • Lines 338-339: please explain
  • Line 341: Effect of bariatric surgery: please add a comment of the role of diet changes after bariatric surgery.
  • Lines 417-424: please add a comment of differences in thosecealiac patients with gluten free diet and with normal diet

Round 2

Reviewer 2 Report

None